# Path-SGD: Path-Normalized Optimization in Deep Neural Networks

**Behnam Neyshabur**
Toyota Technological Institute at Chicago
bneyshabur@ttic.edu

**Ruslan Salakhutdinov**
Departments of Statistics and Computer Science
University of Toronto
rsalakhu@cs.toronto.edu

**Nathan Srebro**
Toyota Technological Institute at Chicago
nati@ttic.edu

## Abstract

We revisit the choice of SGD for training deep neural networks by reconsidering the appropriate geometry in which to optimize the weights. We argue for a geometry invariant to rescaling of weights that does not affect the output of the network, and suggest Path-SGD, which is an approximate steepest descent method with respect to a path-wise regularizer related to max-norm regularization. Path-SGD is easy and efficient to implement and leads to empirical gains over SGD and Ada-Grad.

## 1   Introduction

Training deep networks is a challenging problem [16, 2] and various heuristics and optimization algorithms have been suggested in order to improve the efficiency of the training [5, 9, 4]. However, training deep architectures is still considerably slow and the problem has remained open. Many of the current training methods rely on good initialization and then performing Stochastic Gradient Descent (SGD), sometimes together with an adaptive stepsize or momentum term [16, 1, 6].

Revisiting the choice of gradient descent, we recall that optimization is inherently tied to a choice of geometry or measure of distance, norm or divergence. Gradient descent for example is tied to the $\ell_2$ norm as it is the steepest descent with respect to $\ell_2$ norm in the parameter space, while coordinate descent corresponds to steepest descent with respect to the $\ell_1$ norm and exp-gradient (multiplicative weight) updates is tied to an entropic divergence. Moreover, at least when the objective function is convex, convergence behavior is tied to the corresponding norms or potentials. For example, with gradient descent, or SGD, convergence speeds depend on the $\ell_2$ norm of the optimum. The norm or divergence can be viewed as a regularizer for the updates. There is therefore also a strong link between regularization for optimization and regularization for learning: optimization may provide implicit regularization in terms of its corresponding geometry, and for ideal optimization performance the optimization geometry should be aligned with inductive bias driving the learning [14].

Is the $\ell_2$ geometry on the weights the appropriate geometry for the space of deep networks? Or can we suggest a geometry with more desirable properties that would enable faster optimization and perhaps also better implicit regularization? As suggested above, this question is also linked to the choice of an appropriate regularizer for deep networks.

Focusing on networks with RELU activations, we observe that scaling down the incoming edges to a hidden unit and scaling up the outgoing edges by the same factor yields an equivalent network

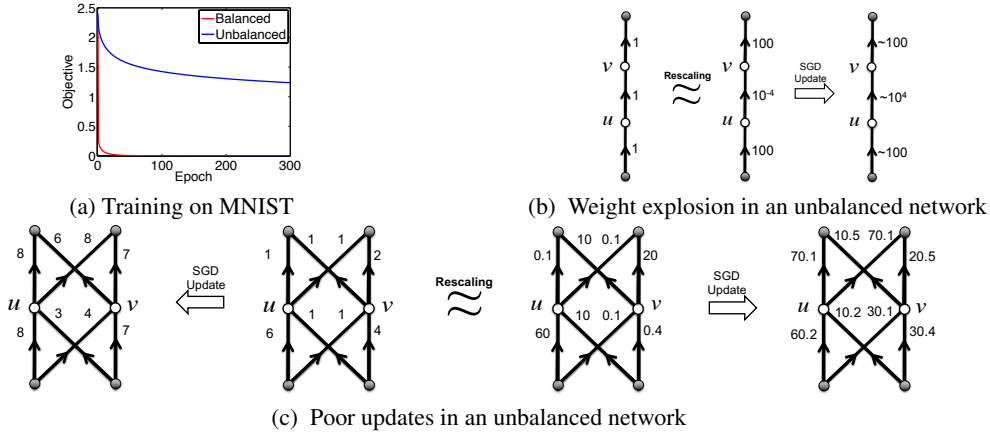

(a) Training on MNIST

(b) Weight explosion in an unbalanced network

(c) Poor updates in an unbalanced network

Figure 1: (a): Evolution of the cross-entropy error function when training a feed-forward network on MNIST with two hidden layers, each containing 4000 hidden units. The unbalanced initialization (blue curve) is generated by applying a sequence of rescaling functions on the balanced initializations (red curve). (b): Updates for a simple case where the input is $x = 1$, thresholds are set to zero (constant), the stepsize is 1, and the gradient with respect to output is $\delta = -1$. (c): Updated network for the case where the input is $x = (1, 1)$, thresholds are set to zero (constant), the stepsize is 1, and the gradient with respect to output is $\delta = (-1, -1)$.

computing the same function. Since predictions are invariant to such rescalings, it is natural to seek a geometry, and corresponding optimization method, that is similarly invariant.

We consider here a geometry inspired by max-norm regularization (regularizing the maximum norm of incoming weights into any unit) which seems to provide a better inductive bias compared to the $\ell_2$ norm (weight decay) [3, 15]. But to achieve rescaling invariance, we use not the max-norm itself, but rather the minimum max-norm over all rescalings of the weights. We discuss how this measure can be expressed as a "path regularizer" and can be computed efficiently.

We therefore suggest a novel optimization method, Path-SGD, that is an approximate steepest descent method with respect to path regularization. Path-SGD is rescaling-invariant and we demonstrate that Path-SGD outperforms gradient descent and AdaGrad for classifications tasks on several benchmark datasets.

**Notations** A feedforward neural network that computes a function $f : \mathbb{R}^D \to \mathbb{R}^C$ can be represented by a directed acyclic graph (DAG) $G(V, E)$ with $D$ input nodes $v_{in}[1], \ldots, v_{in}[D] \in V$, $C$ output nodes $v_{out}[1], \ldots, v_{out}[C] \in V$, weights $w : E \to \mathbb{R}$ and an activation function $\sigma : \mathbb{R} \to \mathbb{R}$ that is applied on the internal nodes (hidden units). We denote the function computed by this network as $f_{G,w,\sigma}$. In this paper we focus on RELU (REctified Linear Unit) activation function $\sigma_{\text{RELU}}(x) = \max\{0, x\}$. We refer to the depth $d$ of the network which is the length of the longest directed path in $G$. For any $0 \leq i \leq d$, we define $V_{\text{in}}^i$ to be the set of vertices with longest path of length $i$ to an input unit and $V_{\text{out}}^i$ is defined similarly for paths to output units. In layered networks $V_{\text{in}}^i = V_{\text{out}}^{d-i}$ is the set of hidden units in a hidden layer $i$.

## 2 Rescaling and Unbalanceness

One of the special properties of RELU activation function is non-negative homogeneity. That is, for any scalar $c \geq 0$ and any $x \in \mathbb{R}$, we have $\sigma_{\text{RELU}}(c \cdot x) = c \cdot \sigma_{\text{RELU}}(x)$. This interesting property allows the network to be rescaled without changing the function computed by the network. We define the *rescaling function* $\rho_{c,v}(w)$, such that given the weights of the network $w : E \to \mathbb{R}$, a constant $c > 0$, and a node $v$, the rescaling function multiplies the incoming edges and divides the outgoing edges of $v$ by $c$. That is, $\rho_{c,v}(w)$ maps $w$ to the weights $\tilde{w}$ for the rescaled network, where for any $(u_1 \to u_2) \in E$:

$$\tilde{w}_{(u_1 \to u_2)} = \begin{cases} c.w_{(u_1 \to u_2)} & u_2 = v, \\ \frac{1}{c} w_{(u_1 \to u_2)} & u_1 = v, \\ w_{(u_1 \to u_2)} & \text{otherwise.} \end{cases} \quad (1)$$

It is easy to see that the rescaled network computes the same function, i.e. $f_{G,w,\sigma_{\text{RELU}}} = f_{G,\rho_{c,v}(w),\sigma_{\text{RELU}}}$. We say that the two networks with weights $w$ and $\tilde{w}$ are *rescaling equivalent* denoted by $w \sim \tilde{w}$ if and only if one of them can be transformed to another by applying a sequence of rescaling functions $\rho_{c,v}$.

Given a training set $\mathcal{S} = \{(x_1, y_n), \ldots, (x_n, y_n)\}$, our goal is to minimize the following objective function:

$$L(w) = \frac{1}{n} \sum_{i=1}^{n} \ell(f_w(x_i), y_i). \tag{2}$$

Let $w^{(t)}$ be the weights at step $t$ of the optimization. We consider update step of the following form $w^{(t+1)} = w^{(t)} + \Delta w^{(t+1)}$. For example, for gradient descent, we have $\Delta w^{(t+1)} = -\eta \nabla L(w^{(t)})$, where $\eta$ is the step-size. In the stochastic setting, such as SGD or mini-batch gradient descent, we calculate the gradient on a small subset of the training set.

Since *rescaling equivalent* networks compute the same function, it is desirable to have an update rule that is not affected by rescaling. We call an optimization method *rescaling invariant* if the updates of rescaling equivalent networks are rescaling equivalent. That is, if we start at either one of the two rescaling equivalent weight vectors $\tilde{w}^{(0)} \sim w^{(0)}$, after applying $t$ update steps separately on $\tilde{w}^{(0)}$ and $w^{(0)}$, they will remain rescaling equivalent and we have $\tilde{w}^{(t)} \sim w^{(t)}$.

Unfortunately, gradient descent is *not* rescaling invariant. The main problem with the gradient updates is that scaling down the weights of an edge will also scale up the gradient which, as we see later, is exactly the opposite of what is expected from a rescaling invariant update.

Furthermore, gradient descent performs very poorly on "unbalanced" networks. We say that a network is *balanced* if the norm of incoming weights to different units are roughly the same or within a small range. For example, Figure 1(a) shows a huge gap in the performance of SGD initialized with a randomly generated balanced network $w^{(0)}$, when training on MNIST, compared to a network initialized with unbalanced weights $\tilde{w}^{(0)}$. Here $\tilde{w}^{(0)}$ is generated by applying a sequence of random rescaling functions on $w^{(0)}$ (and therefore $w^{(0)} \sim \tilde{w}^{(0)}$).

In an unbalanced network, gradient descent updates could blow up the smaller weights, while keeping the larger weights almost unchanged. This is illustrated in Figure 1(b). If this were the only issue, one could scale down all the weights after each update. However, in an unbalanced network, the relative changes in the weights are also very different compared to a balanced network. For example, Figure 1(c) shows how two rescaling equivalent networks could end up computing a very different function after only a single update.

## 3  Magnitude/Scale measures for deep networks

Following [12], we consider the grouping of weights going into each node of the network. This forms the following generic group-norm type regularizer, parametrized by $1 \leq p, q \leq \infty$:

$$\mu_{p,q}(w) = \left( \sum_{v \in V} \left( \sum_{(u \to v) \in E} \left| w_{(u \to v)} \right|^p \right)^{q/p} \right)^{1/q}. \tag{3}$$

Two simple cases of above group-norm are $p = q = 1$ and $p = q = 2$ that correspond to overall $\ell_1$ regularization and weight decay respectively. Another form of regularization that is shown to be very effective in RELU networks is the max-norm regularization, which is the maximum over all units of norm of incoming edge to the unit[1] [3, 15]. The max-norm correspond to "per-unit" regularization when we set $q = \infty$ in equation (4) and can be written in the following form:

$$\mu_{p,\infty}(w) = \sup_{v \in V} \left( \sum_{(u \to v) \in E} \left| w_{(u \to v)} \right|^p \right)^{1/p} \tag{4}$$

Weight decay is probably the most commonly used regularizer. On the other hand, per-unit regularization might not seem ideal as it is very extreme in the sense that the value of regularizer corresponds to the highest value among all nodes. However, the situation is very different for networks with RELU activations (and other activation functions with non-negative homogeneity property). In these cases, per-unit $\ell_2$ regularization has shown to be very effective [15]. The main reason could be because RELU networks can be rebalanced in such a way that all hidden units have the same norm. Hence, per-unit regularization will not be a crude measure anymore.

Since $\mu_{p,\infty}$ is not rescaling invariant and the values of the scale measure are different for rescaling equivalent networks, it is desirable to look for the minimum value of a regularizer among all rescaling equivalent networks. Surprisingly, for a feed-forward network, the minimum $\ell_p$ per-unit regularizer among all rescaling equivalent networks can be efficiently computed by a single forward step. To see this, we consider the vector $\pi(w)$, the *path vector*, where the number of coordinates of $\pi(w)$ is equal to the total number of paths from the input to output units and each coordinate of $\pi(w)$ is the equal to the product of weights along a path from an input nodes to an output node. The $\ell_p$-path regularizer is then defined as the $\ell_p$ norm of $\pi(w)$ [12]:

$$\phi_p(w) = \|\pi(w)\|_p = \left( \sum_{v_{in}[i] \xrightarrow{e_1} v_1 \xrightarrow{e_2} v_2 \ldots \xrightarrow{e_d} v_{out}[j]} \left| \prod_{k=1}^{d} w_{e_k} \right|^p \right)^{1/p} \tag{5}$$

The following Lemma establishes that the $\ell_p$-path regularizer corresponds to the minimum over all equivalent networks of the per-unit $\ell_p$ norm:

**Lemma 3.1** ([12]). $\phi_p(w) = \min_{\tilde{w} \sim w} \left( \mu_{p,\infty}(\tilde{w}) \right)^d$

The definition (5) of the $\ell_p$-path regularizer involves an exponential number of terms. But it can be computed efficiently by dynamic programming in a single forward step using the following equivalent form as nested sums:

$$\phi_p(w) = \left( \sum_{(v_{d-1} \to v_{out}[j]) \in E} \left| w_{(v_{d-1} \to v_{out}[j])} \right|^p \sum_{(v_{d-2} \to v_{d-1}) \in E} \cdots \sum_{(v_{in}[i] \to v_1) \in E} \left| w_{(v_{in}[i] \to v_1)} \right|^p \right)^{1/p}$$

A straightforward consequence of Lemma 3.1 is that the $\ell_p$ path-regularizer $\phi_p$ is invariant to rescaling, i.e. for any $\tilde{w} \sim w$, $\phi_p(\tilde{w}) = \phi_p(w)$.

## 4 Path-SGD: An Approximate Path-Regularized Steepest Descent

Motivated by empirical performance of max-norm regularization and the fact that path-regularizer is invariant to rescaling, we are interested in deriving the steepest descent direction with respect to the path regularizer $\phi_p(w)$:

$$w^{(t+1)} = \arg\min_{w} \ \eta \left\langle \nabla L(w^{(t)}), w \right\rangle + \frac{1}{2} \left\| \pi(w) - \pi(w^{(t)}) \right\|_p^2 \tag{6}$$

$$= \arg\min_{w} \ \eta \left\langle \nabla L(w^{(t)}), w \right\rangle + \frac{1}{2} \left( \sum_{v_{in}[i] \xrightarrow{e_1} v_1 \xrightarrow{e_2} v_2 \ldots \xrightarrow{e_d} v_{out}[j]} \left| \prod_{k=1}^{d} w_{e_k} - \prod_{k=1}^{d} w_{e_k}^{(t)} \right|^p \right)^{2/p}$$

$$= \arg\min_{w} J^{(t)}(w)$$

The steepest descent step (6) is hard to calculate exactly. Instead, we will update each coordinate $w_e$ independently (and synchronously) based on (6). That is:

$$w_e^{(t+1)} = \arg\min_{w_e} \ J^{(t)}(w) \qquad \text{s.t. } \forall_{e' \neq e} \ w_{e'} = w_{e'}^{(t)} \tag{7}$$

Taking the partial derivative with respect to $w_e$ and setting it to zero we obtain:

$$0 = \eta \frac{\partial L}{\partial w_e}(w^{(t)}) + \left( w_e - w_e^{(t)} \right) \left( \sum_{v_{in}[i] \ldots \xrightarrow{e} \ldots v_{out}[j]} \prod_{e' \neq e} \left| w_{e'}^{(t)} \right|^p \right)^{2/p}$$

---

**Algorithm 1** Path-SGDupdate rule

---

1: $\forall_{v \in V_{\text{in}}^0}\ \gamma_{\text{in}}(v) = 1$                                                                      ▷ Initialization
2: $\forall_{v \in V_{\text{out}}^0}\ \gamma_{\text{out}}(v) = 1$
3: **for** $i = 1$ **to** $d$ **do**
4:      $\forall_{v \in V_{\text{in}}^i} \gamma_{\text{in}}(v) = \sum_{(u \to v) \in E} \gamma_{\text{in}}(u) \left| w_{(u,v)} \right|^p$
5:      $\forall_{v \in V_{\text{out}}^i} \gamma_{\text{out}}(v) = \sum_{(v \to u) \in E} \left| w_{(v,u)} \right|^p \gamma_{\text{out}}(u)$
6: **end for**
7: $\forall_{(u \to v) \in E}\ \gamma(w^{(t)}, (u, v)) = \gamma_{\text{in}}(u)^{2/p} \gamma_{\text{out}}(v)^{2/p}$
8: $\forall_{e \in E} w_e^{(t+1)} = w_e^{(t)} - \frac{\eta}{\gamma(w^{(t)}, e)} \frac{\partial L}{\partial w_e}(w^{(t)})$                            ▷ Update Rule

---

where $v_{\text{in}}[i] \cdots \xrightarrow{e} \ldots v_{\text{out}}[j]$ denotes the paths from any input unit $i$ to any output unit $j$ that includes $e$. Solving for $w_e$ gives us the following update rule:

$$\hat{w}_e^{(t+1)} = w_e^{(t)} - \frac{\eta}{\gamma_p(w^{(t)}, e)} \frac{\partial L}{\partial w}(w^{(t)}) \tag{8}$$

where $\gamma_p(w, e)$ is given as

$$\gamma_p(w, e) = \left( \sum_{v_{\text{in}}[i] \cdots \xrightarrow{e} \ldots v_{\text{out}}[j]} \prod_{e' \neq e} |w_{e'}|^p \right)^{2/p} \tag{9}$$

We call the optimization using the update rule (8) path-normalized gradient descent. When used in stochastic settings, we refer to it as Path-SGD.

Now that we know Path-SGDis an approximate steepest descent with respect to the path-regularizer, we can ask whether or not this makes Path-SGDa *rescaling invariant* optimization method. The next theorem proves that Path-SGDis indeed rescaling invariant.

**Theorem 4.1.** *Path-SGDis rescaling invariant.*

*Proof.* It is sufficient to prove that using the update rule (8), for any $c > 0$ and any $v \in E$, if $\tilde{w}^{(t)} = \rho_{c,v}(w^{(t)})$, then $\tilde{w}^{(t+1)} = \rho_{c,v}(w^{(t+1)})$. For any edge $e$ in the network, if $e$ is neither incoming nor outgoing edge of the node $v$, then $\tilde{w}(e) = w(e)$, and since the gradient is also the same for edge $e$ we have $\tilde{w}_e^{(t+1)} = w_e^{(t+1)}$. However, if $e$ is an incoming edge to $v$, we have that $\tilde{w}^{(t)}(e) = cw^{(t)}(e)$. Moreover, since the outgoing edges of $v$ are divided by $c$, we get $\gamma_p(\tilde{w}^{(t)}, e) = \frac{\gamma_p(w^{(t)}, e)}{c^2}$ and $\frac{\partial L}{\partial w_e}(\tilde{w}^{(t)}) = \frac{\partial L}{c \partial w_e}(w^{(t)})$. Therefore,

$$\tilde{w}_e^{(t+1)} = cw_e^{(t)} - \frac{c^2 \eta}{\gamma_p(w^{(t)}, e)} \frac{\partial L}{c \partial w_e}(w^{(t)})$$

$$= c \left( w^{(t)} - \frac{\eta}{\gamma_p(w^{(t)}, e)} \frac{\partial L}{\partial w_e}(w^{(t)}) \right) = cw_e^{(t+1)}.$$

A similar argument proves the invariance of Path-SGDupdate rule for outgoing edges of $v$. Therefore, Path-SGDis rescaling invariant. ◻

**Efficient Implementation:** The Path-SGD update rule (8), in the way it is written, needs to consider all the paths, which is exponential in the depth of the network. However, it can be calculated in a time that is no more than a forward-backward step on a single data point. That is, in a mini-batch setting with batch size $B$, if the backpropagation on the mini-batch can be done in time $BT$, the running time of the Path-SGD on the mini-batch will be roughly $(B + 1)T$ – a very moderate runtime increase with typical mini-batch sizes of hundreds or thousands of points. Algorithm 1 shows an efficient implementation of the Path-SGD update rule.

We next compare Path-SGDto other optimization methods in both balanced and unbalanced settings.

Table 1: General information on datasets used in the experiments.

| Data Set | Dimensionality | Classes | Training Set | Test Set |
|----------|----------------|---------|--------------|----------|
| CIFAR-10 | 3072 ($32 \times 32$ color) | 10 | 50000 | 10000 |
| CIFAR-100 | 3072 ($32 \times 32$ color) | 100 | 50000 | 10000 |
| MNIST | 784 ($28 \times 28$ grayscale) | 10 | 60000 | 10000 |
| SVHN | 3072 ($32 \times 32$ color) | 10 | 73257 | 26032 |

## 5 Experiments

In this section, we compare $\ell_2$-Path-SGDto two commonly used optimization methods in deep learning, SGD and AdaGrad. We conduct our experiments on four common benchmark datasets: the standard MNIST dataset of handwritten digits [8]; CIFAR-10 and CIFAR-100 datasets of tiny images of natural scenes [7]; and Street View House Numbers (SVHN) dataset containing color images of house numbers collected by Google Street View [10]. Details of the datasets are shown in Table 1.

In all of our experiments, we trained feed-forward networks with two hidden layers, each containing 4000 hidden units. We used mini-batches of size 100 and the step-size of $10^{-\alpha}$, where $\alpha$ is an integer between 0 and 10. To choose $\alpha$, for each dataset, we considered the validation errors over the validation set (10000 randomly chosen points that are kept out during the initial training) and picked the one that reaches the minimum error faster. We then trained the network over the entire training set. All the networks were trained both with and without dropout. When training with dropout, at each update step, we retained each unit with probability 0.5.

We tried both balanced and unbalanced initializations. In balanced initialization, incoming weights to each unit $v$ are initialized to i.i.d samples from a Gaussian distribution with standard deviation $1/\sqrt{\text{fan-in}(v)}$. In the unbalanced setting, we first initialized the weights to be the same as the balanced weights. We then picked 2000 hidden units randomly with replacement. For each unit, we multiplied its incoming edge and divided its outgoing edge by $10c$, where $c$ was chosen randomly from log-normal distribution.

The optimization results without dropout are shown in Figure 2. For each of the four datasets, the plots for objective function (cross-entropy), the training error and the test error are shown from left to right where in each plot the values are reported on different epochs during the optimization. Although we proved that Path-SGDupdates are the same for balanced and unbalanced initializations, to verify that despite numerical issues they are indeed identical, we trained Path-SGDwith both balanced and unbalanced initializations. Since the curves were exactly the same we only show a single curve.

We can see that as expected, the unbalanced initialization considerably hurts the performance of SGD and AdaGrad (in many cases their training and test errors are not even in the range of the plot to be displayed), while Path-SGDperforms essentially the same. Another interesting observation is that even in the balanced settings, not only does Path-SGDoften get to the same value of objective function, training and test error faster, but also the final generalization error for Path-SGDis sometimes considerably lower than SGD and AdaGrad (except CIFAR-100 where the generalization error for SGD is slightly better compared to Path-SGD). The plots for test errors could also imply that implicit regularization due to steepest descent with respect to path-regularizer leads to a solution that generalizes better. This view is similar to observations in [11] on the role of implicit regularization in deep learning.

The results for training with dropout are shown in Figure 3, where here we suppressed the (very poor) results on unbalanced initializations. We observe that except for MNIST, Path-SGDconvergences much faster than SGD or AdaGrad. It also generalizes better to the test set, which again shows the effectiveness of path-normalized updates.

The results suggest that Path-SGDoutperforms SGD and AdaGrad in two different ways. First, it can achieve the same accuracy much faster and second, the implicit regularization by Path-SGDleads to a local minima that can generalize better even when the training error is zero. This can be better analyzed by looking at the plots for more number of epochs which we have provided in the supplementary material. We should also point that Path-SGD can be easily combined with AdaGrad to take

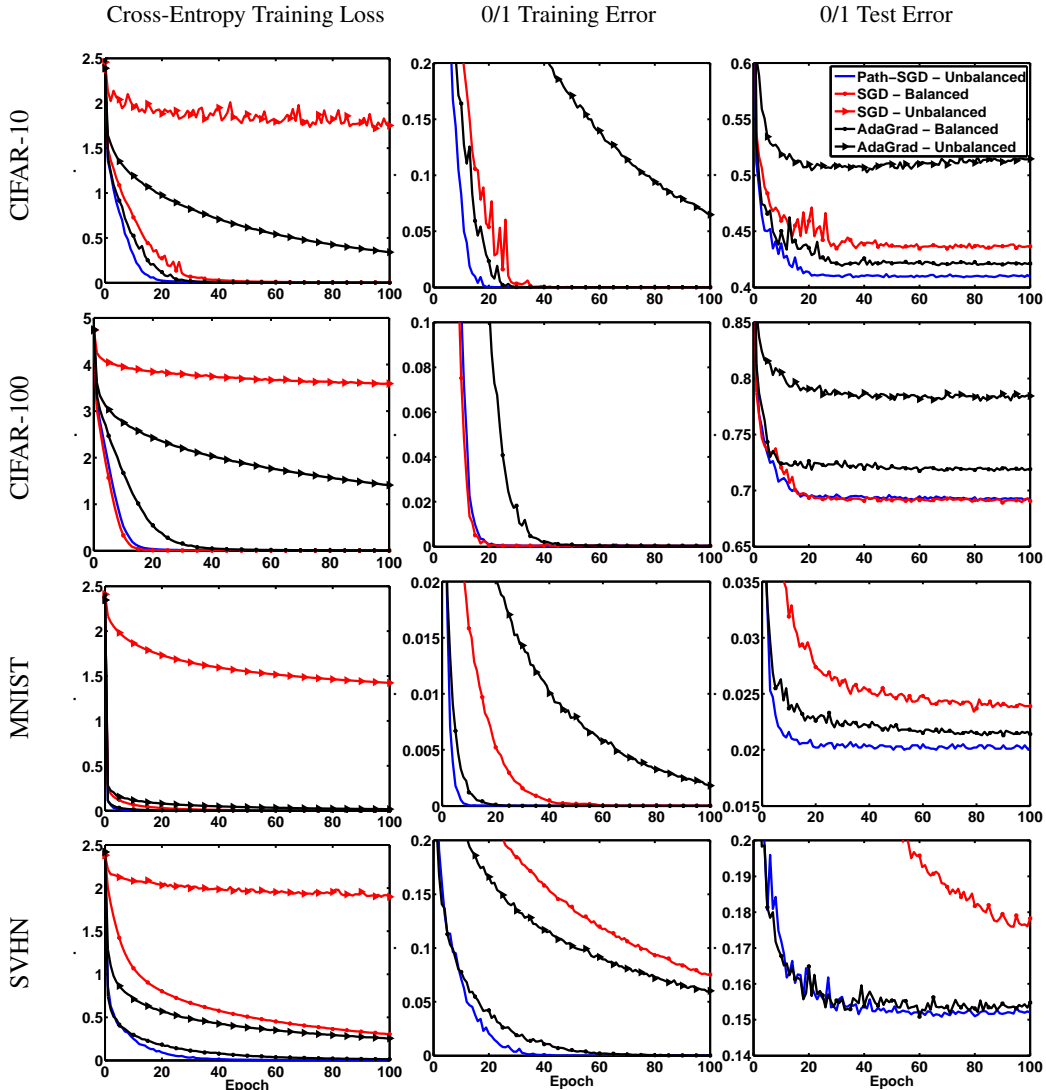

Figure 2: Learning curves using different optimization methods for 4 datasets without dropout. Left panel displays the cross-entropy objective function; middle and right panels show the corresponding values of the training and test errors, where the values are reported on different epochs during the course of optimization. Best viewed in color.

advantage of the adaptive stepsize or used together with a momentum term. This could potentially perform even better compare to Path-SGD.

## 6   Discussion

We revisited the choice of the Euclidean geometry on the weights of RELU networks, suggested an alternative optimization method approximately corresponding to a different geometry, and showed that using such an alternative geometry can be beneficial. In this work we show proof-of-concept success, and we expect Path-SGD to be beneficial also in large-scale training for very deep convolutional networks. Combining Path-SGD with AdaGrad, with momentum or with other optimization heuristics might further enhance results.

Although we do believe Path-SGD is a very good optimization method, and is an easy plug-in for SGD, we hope this work will also inspire others to consider other geometries, other regularizers and perhaps better, update rules. A particular property of Path-SGD is its rescaling invariance, which we

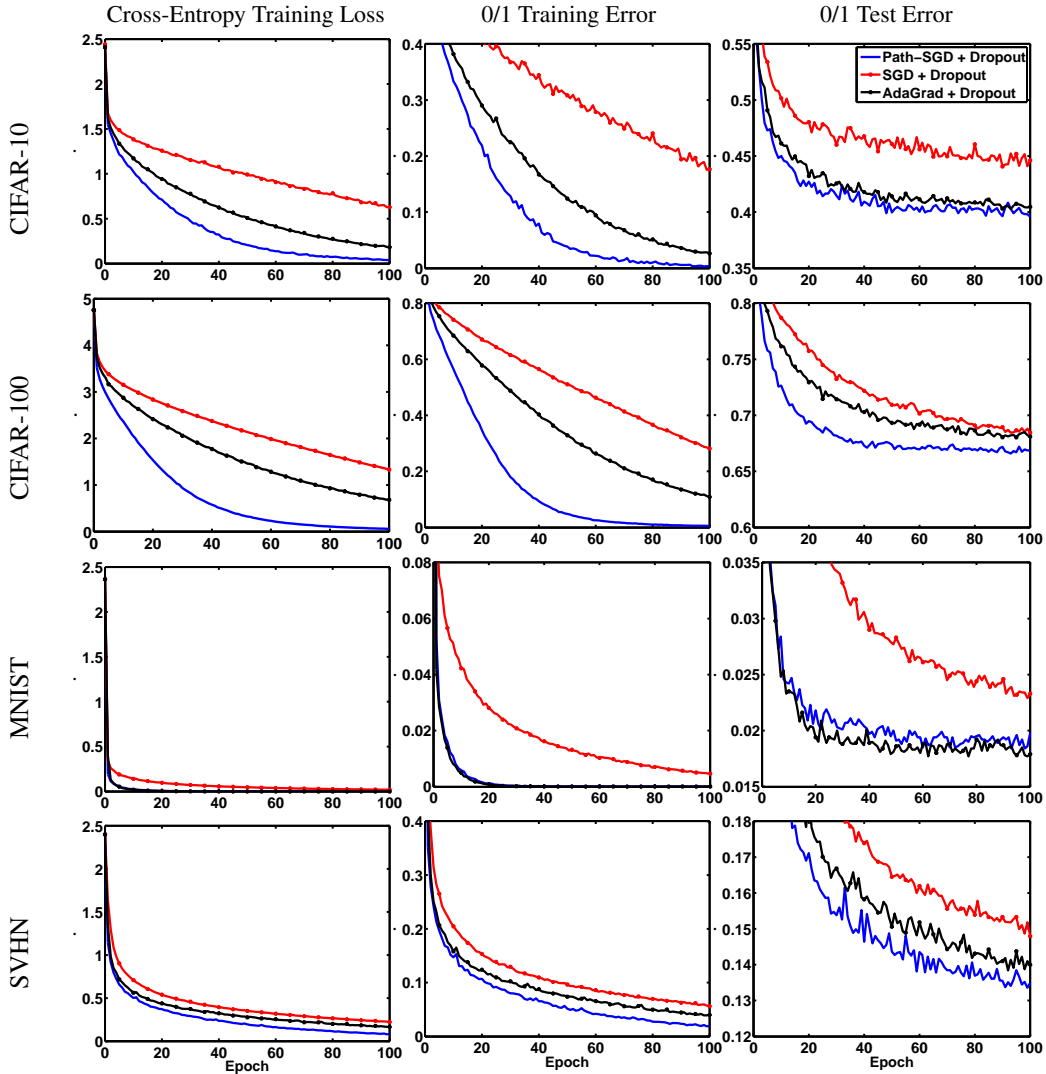

Figure 3: Learning curves using different optimization methods for 4 datasets with dropout. Left panel displays the cross-entropy objective function; middle and right panels show the corresponding values of the training and test errors. Best viewed in color.

argue is appropriate for RELU networks. But Path-SGD is certainly not the only rescaling invariant update possible, and other invariant geometries might be even better.

Path-SGD can also be viewed as a tractable approximation to natural gradient, which ignores the activations, the input distribution and dependencies between different paths. Natural gradient updates are also invariant to rebalancing but are generally computationally intractable.

Finally, we choose to use steepest descent because of its simplicity of implementation. A better choice might be mirror descent with respect to an appropriate potential function, but such a construction seems particularly challenging considering the non-convexity of neural networks.

### Acknowledgments

Research was partially funded by NSF award IIS-1302662 and Intel ICRI-CI. We thank Ryota Tomioka and Hao Tang for insightful discussions and Leon Bottou for pointing out the connection to natural gradient.

## Footnotes

[1] This definition of max-norm is a bit different than the one used in the context of matrix factorization [13]. The later is similar to the minimum upper bound over $\ell_2$ norm of both outgoing edges from the input units and incoming edges to the output units in a two layer feed-forward network.

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
