[Reviews · NeurIPS 2015]

Submitted by Assigned_Reviewer_1

Deep rectified neural networks are over-parameterized in the sense that scaling of the weights in one layer, can be compensated for exactly in the subsequent layer. This paper introduces Path-SGD, a simple modification to the SGD update rule, whose update is invariant to such rescaling. The method is derived from the proximal form of gradient descent, whereby a constraint term is added which preserves the norm of the "product weight" formed along each path in the network (from input to output node). Path-SGD is thus principled and shown to yield faster convergence for a standard 2 layer rectifier network, across a variety of dataset (MNIST, CIFAR-10, CIFAR-100, SVHN). As the method implicitly regularizes the neural weights, this also translates to better generalization performance on half of the datasets.

As an algorithm, Path-SGD appears effective, simple to implement and addresses an obvious flaw in first-order updates to ReLU networks. As a paper, it could be improved however, especially with respect to notation (see details section below) which sometimes obfuscates simple concepts. Certain sections of the paper seem rushed and would require a careful rewrite. See details section below.

There are also a number of glaring omissions to prior work. At its core, Path-SGD belongs to the family of learning algorithms which aim to be invariant to model reparametrizations. This is the central tenet of Amari's natural gradient (NG), whose importance has resurfaced in the area of deep learning (see e.g [R1-R4]). Path-SGD can thus be cast an approximation to NG, which focuses on a particular type of rescaling between neighboring layers. The paper would greatly benefit from such a discussion in my opinion. I also believe NG to be a much more direct way to motivate Path-SGD, than the heuristics of max-norm regularization.

The experimental section could also benefit from a few extra experiments. Can the authors validate experimentally that $\pi(w)$ is more stable during optimization ? It is also regrettable that the authors chose a fixed model architecture for the experiments. One would expect the advantage of Path-SGD to be even more pronounced for deeper networks, which have longer paths from input to output. Such experiments would be much more informative that the dropout experiments of Figure 3, which do not add much to the narrative.

Detailed Feedback: Theorems should be rather self contained and mathematically precise. Path-SGD is not invariant to arbitrary rescaling, contrary to the claims of Theorem 4.1, but only invariant to the rescaling function $\rho$ for c > 0. Please correct accordingly. Eq. 5 (and most others) are missing an outer summation over i and j. Eq. 5: notation $v_in[i] \rightarrow ... \rightarrow v_out[j]$ found throughout the paper is quite cumbersome. One could instead use $\mathcal{P}$ to denote the set of all possible paths in the network, and then for each path $p \in \mathcal{P}$, sum over edges $\e_k \in p$ ? Idem for line 198 and other equation involving paths. line 212: missing $\mathcal{1}{2}$ term line 212: missing absolute values before exponentiating by $p$. line 216: "hard to calculate". Please be more precise: is it intractable, ill-defined, expensive to compute ? line 216: "Instead, we will update each coordinate" -> "perform coordinate descent". The original form was confusing to me as SGD also updates each parameter "independently", in the sense that they do not exploit curvature / covariance information. Eq 7 vs line 224: inconsistent notation between $e'$ and $e_k$ line 227: similar to previous comments, it is clumsy for $v_in[i] -> ... e .. -> v_out[j]$ to be implicitly defined over all i and j.

[R1] Enhanced Gradient for Training Restricted Boltzmann Machines, KyungHyun Cho, Tapani Raiko, Alexander Ilin. ICML 2011. [R2] Deep Learning Made Easier by Linear Transformations in Perceptrons. Tapani Raiko Harri Valpola Yann LeCun. AISTATS'12. [R3] Optimizing Neural Networks with Kronecker-factored Approximate Curvature, James Martens, Roger Grosse. ICML 2015. [R4] Scaling up Natural Gradient by Sparsely Factorizing the Inverse Fisher Matrix. Roger Grosse, Ruslan Salakhudinov. ICML 2015.
Summary: A simple extension to SGD which accounts for the scale over-parametrization in ReLU networks. The algorithm appears effective and simple to implement, and is properly validated on 3 benchmark datasets. A few notation issues hamper the readability of the paper, and the paper could benefit from extra discussions relating Path-SGD to Natural Gradient methods.

Submitted by Assigned_Reviewer_2

The authors propose a stochastic gradient algorithm which is invariant to equivalent reparametrizations of a feedforward network with ReLU activations.

The issue of making SGD invariant to reparametrizations is an important one which has been addressed many times in the literature, though not of these techniques gained wide acceptance for training deep nets. One of the main reasons is that they rely on full matrix inversions, which are intractable when the number of parameters is large. One of the pioneering works is that of Amari: Natural Gradient Works Efficiently in Learning. Since then, there have been other attempts, such that of Martens (cited) or Le Roux et al. (Topmoumoute online natural gradient algorithm).

It would have been interesting to compare to such methods (though the TONGA work has too many hyperparameters to be easily implemented) rather than just SGD and AdaGrad.

Further, as with the other methods mentioned, it seems that the update cost of Path-SGD is larger than that of SGD or AdaGrad. However, the experimental results seem to use the number of epochs as the x-axis, which unduly favors Path-SGD.

In conclusion, this is a very interesting line of work but I feel this paper lacks the proper comparisons and experimental results to grant publication. However, I would be very interested in reading a more refined version of this work.
Summary: The idea is interesting but seems to suffer from the same computational inefficiencies than other similar works, which are not mentioned. I am willing to revise my score if the authors can convince me that their method is also computationally efficient.

Submitted by Assigned_Reviewer_3

The paper introduces a new variant of SGD for ReLU networks, called Path-SGD, which (to my understanding) essentially computes a scaling factor on the derivative for each weight in the network, that depends on the other weights in the network-- the basic idea being that if a weight is going to get multiplied by large weights elsewhere in the network, we should update it more slowly than we otherwise might.

Their method makes the update invariant to rescalings of layers or weights that would normally make SGD behave badly (while maintaining equivalence as models).

I think it's an interesting idea, and in an interesting space of ideas.

Personally I think there is more promise in data-dependent ways to get this type of invariance-- things like natural gradient-- but the formulation of this is nice.

Summary: Interesting and novel idea.

Submitted by Assigned_Reviewer_4

The paper presents Path-SGD, an approximate steepest descent method with respect to a path-wise regularizer (max-norm in the paper).

It is very well motivated from an optimization point of view

and easy to follow.

However, the choice of rescaling equivalent networks could

seem a bit artificial. Why such property and not something else? There are several other ways to plug it in the network architecture; for example one could use

batch-normalization and mitigate such effect.

Experiments are performed on a two hidden layer network. Why not a deeper one? There one could really check the impact of the proposed Path-SGD method when optimization starts to become difficult. What about comparisons with RMSprop and Adadelta for example? What if the network uses batch-normalization?

Not surprisingly the gap between balanced and unbalanced networks is huge, Path-SGD is designed to work well in such situation.

Can you provide a situation where it does not work as well? What kind of properties in the network break it? However, when the network is properly initialized the margin becomes slim (test set).

What about other path-wise regularizers?

Will the code be released?

Overall I think it is a very interesting approach that is worth investigating. The paper in the current format requires some more polishing and additional comparisons.

Summary: The paper is well written and tackles very interesting research topic. More details on the choice of rescaling invariance and more comparisons in the experiments should be added.

Submitted by Assigned_Reviewer_5

The authors propose a new optimization method for neural nets using ReLU units. Learning rates in SGD are set in such a way that the optimization becomes stable w.r.t. rescaling of the weight matrices. Each learning rate is inverse proportional to the group norm of all the paths that include the weight under consideration (and this can be computed efficiently). Results are reported on MNIST, CIFAR and SVHN datasets.

Quality The overall quality is ok. The weakest part is the empirical validation. Improvements are generally very marginal compared to vanilla SGD and Adagrad and they are not easy to assess. First, the authors should have reported test error VS time as opposed to number of epochs. Second, the absolute performance reported is not very good - probably because a fully connected as opposed to convolutional network has been considered. Third, stronger baselines are needed (SGD with learning rate annealing, SGD with momentum).

Clarity Pretty good.

Originality Limited: most of the theoretical work has been presented in [12]. The novel part is algorithm 1 which is barely explained.

Significance Low: since pretty much everybody nowadays works with convolutional neural networks and not fully connected nets. The empirical results probably won't convince the community to try this out.

Overall suggestion: strengthen the empirical validation as recommended above and adapt this to CNNs.

I have read the other reviews and the rebuttal.

Overall, I think my initial rating was too negative and I am raising it. However, I still think that this paper is not ready for publication. My main concern is still that reporting test error versus time is a must when proposing a new optimization method. The fact that certain parts of the proposed method are less parallelizable is a useful fact to know. Factoring this in, it is still not clear to me whether the proposed method is useful in practice. I am afraid that the paper would be a bit misleading without this important piece of information.
Summary: The empirical validation is weak (both in terms of the results and in terms of the protocol used) and the proposed method is applied to fully connected nets which are not used very much these days.

Author Feedback
Author rebuttal: We thank all the reviewers for their valuable comments. We will incorporate the feedback in the final version, including notation and consistency as well as release the code for Path-SGD.

1- Connection to Natural Gradient: We realized this very interesting connection only after the submission. Path-SGD can be viewed as a tractable data-independent approximation of the natural gradient where (1) the marginal distribution over the inputs is ignored; (2) the correlation between the events that two different paths are "active" is ignored. That is, Path-SGD scalings can be derived to be an approximation of the diagonal of the Fisher information matrix in Natural Gradient. We plan on adding a discussion of this relationship that indeed helps better understand and motivate Path-SGD, understand its relationship to other methods that can be viewed as different approximation to the Natural Gradient (eg TONGA and Batch Normalization), as well as suggests ways of improving Path-SGD while still maintaining tractability.

2- Computational Efficiency: We have discussed computational efficiency in lines 262-269. In the mini-batch setting, each epoch of Path-SGD is about 1 percent slower compared to the mini-batch gradient descent when the batch-size is 100 (or 0.1 percent slower if the batch-size is 1000). This is because Path-SGD requires one additional forward pass for each mini-batch (See Algorithm 1).

In our unoptimized implementation, Path-SGD with mini-batch of size 100 is about 5 percent slower than mini-batch gradient descent (5% and not 1% because the additional forward propagation does not use the GPUs as effectively as the rest of the mini-batch). This is still a fairly minor difference, which can probably be reduced to 1% via GPU usage optimization, and and so we still think comparing iterations rather than runtime is more informative, especially since the experiments are there to demonstrate an idea, not compete in a competition. Path-SGD is also better candidate for optimization because it often achieves a better final test error.

The code for training our models and reproducing our experimental results will be made available online.

3- Experiments with deep Convnet models: We are currently running several experiments on very deep networks (more than 10 layers) as well as on deep ConvNet models. We will include these experiments in the final version.

4- Use of ConvNets: We disagree with Rev 4 statement that pretty much everybody just works with ConvNets. ConvNets are heavily used in learning image features, but for many other domains, including speech recognition, NLP, including probabilistic neural language models,, one does not use ConvNets.

5- Comparisons to other baselines: We compared Path-SGD with SGD and AdaGrad because these methods are widely used to train state-of-the-art deep neural nets. The results of the Path-SGD can be easily enhanced by combining it with momentum term, or batch normalization, or even AdaGrad updates. Our main focus in this work was to compare the Path-SGD with SGD updates, as SGD is still the mostly commonly used technique for training deep neural networks.